# Tidal influence on carbon dioxide and methane fluxes from tree stems and soils in mangrove forests

Zhao-Jun Yong[1], Wei-Jen Lin[1,2], Chiao-Wen Lin[2], Hsing-Juh Lin[1*]

[1]Department of Life Sciences and Innovation and Development Center of Sustainable Agriculture, National Chung Hsing University, Taichung 40227, Taiwan

[2]Department of Marine Environment and Engineering and The Center for Water Resources Studies, National Sun Yat-sen University, Kaohsiung 80424, Taiwan

*Correspondence to*: Hsing-Juh Lin (hjlin@dragon.nchu.edu.tw)

**Abstract.** Mangroves are critical blue carbon ecosystems. Measurements of methane ($CH_4$) emissions from mangrove tree stems have the potential to reduce the uncertainty in the capacity of carbon sequestration. This study is the first to simultaneously measure the $CH_4$ fluxes from both stems and soils throughout tidal cycles. We quantified carbon dioxide ($CO_2$) and $CH_4$ fluxes from mangrove tree stems of *Avicennia marina* and *Kandelia obovata* during tidal cycles, which have distinct root structures. The mangrove tree stems served as both net $CO_2$ and $CH_4$ sources. Compared to those of the soils, the mangrove tree stems exhibited remarkably lower $CH_4$ fluxes, but no difference in $CO_2$ fluxes. The stems of *A. marina* exhibited an increasing trend in the $CO_2$ flux from low to high tides. On the other hand, $CH_4$ flux showed high temporal variability, with the stems of *A. marina* functioning as a $CH_4$ sink before tidal inundation and becoming a source after ebbing. In contrast, the stems of *K. obovata* showed no consistent pattern of the $CO_2$ or $CH_4$ flux. Based on our findings, the stem $CH_4$ fluxes of *A. marina* could vary by up to 1200% when considering tidal influence, compared to ignoring tidal influence. Therefore, sampling only during low tides might underestimate the stem $CO_2$ and $CH_4$ fluxes on a diurnal scale. This study highlights the necessity of considering tidal influence when quantifying GHG fluxes from mangrove tree stems. Further research is needed to explore the underlying mechanisms driving the observed flux variations and improve the understanding of GHG dynamics in mangrove ecosystems.

## 1 Introduction

Global methane ($CH_4$) emissions have reached a record high level (Saunois et al., 2020). Currently, there are two primary methods utilized for assessing global $CH_4$ emissions: the bottom-up method and the top-down method. The bottom-up method relies on compiling data from greenhouse gas (GHG) inventories and biogeochemical models to infer the sources of emissions. On the other hand, the top-down method involves measuring atmospheric $CH_4$ concentrations and utilizing transport models to infer the sources of emissions in order to estimate and assess $CH_4$ emissions on a global scale. $CH_4$ emissions estimated by the bottom-up method are significantly higher than those estimated by the top-down method, indicating a high degree of uncertainty and suggesting that some sources may be overlooked or not well understood (Jackson et al., 2020). $CH_4$ generated in wetlands can be released into the atmosphere not only through diffusion, ebullition, and transport mediated by herbaceous

plants but also through the stems of woody plants (Gauci et al., 2010; Terazawa et al., 2007). Pangala et al. (2017) demonstrated that the difference between the top-down and bottom-up estimates of $CH_4$ emissions could be accounted for by the upscaled $CH_4$ flux from tree stems, emphasizing the necessity of considering this pathway in carbon budgets (Carmichael et al., 2014).

Furthermore, forest wetlands account for approximately 60% of the global wetland area, highlighting the potential contribution of woody stems to the global GHG emissions (Barba et al., 2019a; Covey and Megonigal, 2019). While carbon dioxide ($CO_2$) exchange at the stem–atmosphere interface has been examined (Teskey et al., 2008), little is known regarding the sources and mechanisms of $CH_4$ emissions originating from tree stems relative to those originating from other pathways. $CH_4$ emitted by tree stems may originate from microorganisms or cryptogams within the stem bark (Jeffrey et al., 2021; Lenhart et al., 2015)

or from the soil, where it is produced and enters the roots before being transported in either liquid or gaseous form through xylem or aerenchyma tissue (Kutschera et al., 2016; Vroom et al., 2022).

GHG emissions from tree stems exhibit temporal and spatial variations with different influencing mechanisms in various studies: i) the tree stem GHG flux tends to be higher during the growing season and lower during the dormant season, but there may also be no significant differences among seasons (Barba et al., 2019b; Köhn et al., 2021; Pangala et al., 2015; Pitz et al.,

2018; Wang et al., 2016; Zhang et al., 2022); ii) significant variations in the GHG fluxes from tree stems have been observed at different heights above ground level, with a decreasing trend along the tree stem height (Moldaschl et al., 2021; Pangala et al., 2013, 2014, 2015; Sjögersten et al., 2020), although some studies have not reported this phenomenon (Machacova et al., 2021; Wang et al., 2016); iii) the tree stem GHG emissions may be regulated by various environmental factors such as temperature, moisture, and redox potential (Barba et al., 2019b; Gao et al., 2021; Jeffrey et al., 2019; Pitz et al., 2018; Schindler

et al., 2020, 2021; Sjögersten et al., 2020; Terazawa et al., 2015), which can be affected by the fluctuations of water table height due to seasonal changes and hydrological processes (Jeffrey et al., 2023; Peacock et al., 2024; Terazawa et al., 2021); iv) tree physiological factors such as lenticel density, wood density, water content, and stem bark structure may also influence the GHG fluxes originating from tree stems (Jeffrey et al., 2024; Pangala et al., 2013, 2014, 2015; Wang et al., 2016; Zhang et al., 2022).

However, most related studies have focused on freshwater wetlands and upland forests, while relatively limited research has focused on mangrove forests. Jeffrey et al. (2019) reported that dead mangrove trees may contribute approximately 26% to the $CH_4$ emissions in mangrove ecosystems. However, He et al. (2019) reported inconsistent results, revealing a relatively small contribution from tree stems. The contribution of mangrove tree stems to the total GHG flux in ecosystems is generally less than that in soil (Gao et al., 2021; He et al., 2019; Jeffrey et al., 2019) but still has the potential to exceed 50% (Zhang et al.,

2022). Additionally, the GHG fluxes from mangrove tree stems vary among tree species (Zhang et al., 2022) and may even differ within a single tree species (Gao et al., 2021), highlighting the uncertainty in the GHG emissions from mangrove tree stems and emphasizing the need for further investigation.

Mangroves are primarily distributed in tropical and subtropical coastal regions and are regarded as critical ecosystems with a high capacity for sequestering blue carbon (Li et al., 2018; Duarte de Paula Costa and Macreadie, 2022). The anaerobic

conditions resulting from tidal inundation, along with the abundant organic matter, turn mangrove soil into a source of $CH_4$

emissions (Lin et al., 2020). This, in turn, impacts their role in mitigating global warming. Moreover, several studies have demonstrated the influence of tides on the emission of GHGs in coastal wetlands. In both seagrass meadows and tidal marshes, the $CH_4$ flux tends to peak when tidal water reaches the sampling site (Bahlmann et al., 2015; Capooci and Vargas, 2022). The sudden release of $CH_4$ can occur through physical force under the influence of tidal movement (Li et al., 2021), resulting in

the advective exchange of groundwater or soil pore water with the overlying surface water (Billerbeck et al., 2006; Rosentreter et al., 2018). $CH_4$ emissions during tidal inundation may be higher if tidal water contains high concentrations of dissolved $CH_4$, which can increase the emissions of $CH_4$ through diffusion due to the concentration gradient (Sturm et al., 2017; Tong et al., 2013). Yamamoto et al. (2009) reported a positive correlation between the water table and GHG fluxes in the flooded littoral zone with vegetation, suggesting that the water pressure rather than gas diffusion primarily affects the emissions of $CO_2$ and

$CH_4$ across the water–atmosphere interface by ejecting gases from pore spaces. This finding is contrary to previous results in which lower $CH_4$ fluxes were observed during high tide, which may be caused by the higher water pressure limiting $CH_4$ diffusion in soil pore spaces filled with water and plant-mediated transport (Tong et al., 2010; Tong et al., 2013). Additionally, $CH_4$ may be oxidized during diffusion in water (Tong et al., 2013). Furthermore, if the dissolved oxygen concentration, sulfate concentration, and salinity are high in tidewater, this may inhibit $CH_4$ production and/or promote $CH_4$ oxidation (Huang et al.,

2019), resulting in lower $CH_4$ emissions during high tides. The variation in the $CH_4$ flux across the water–atmosphere interface during tidal inundation could be driven by current or wind-induced turbulence (Sturm et al., 2017). $CH_4$ emissions even exhibited different trends during spring and neap tides (Huang et al., 2019; Tong et al., 2013). However, to our knowledge, there is only one study on the GHG fluxes from mangrove tree stems during tidal cycles (Epron et al., 2023).

This study aimed to quantify the $CO_2$ and $CH_4$ emissions from the tree stems of *K. obovata* and *A. mariana*, which are the

dominant mangrove species with distinct root structures distributed on the northern and southern coasts of Taiwan, respectively. We investigated the temporal variations in the stem GHG fluxes during tidal cycles and assessed the influence of tides on the upscaled flux. We also simultaneously measured the GHG emissions from mangrove soil, even during tidal inundation, to compare the temporal dynamics of GHG fluxes between the tree stems and soil. We hypothesized that the GHG fluxes from mangrove tree stems and soil exhibit synchronized temporal and species variation during the tidal cycle and that the tidal cycle

may exert a significant impact on GHG emissions on a larger scale.

## 2 Materials and Methods

### 2.1 Site description

This study focused on the mangroves at four sites along the western coast of Taiwan (Fig. 1). The dominant mangrove species in Wazihwei (K-WZW; 25°10′N, 121°25′E) and Xinfeng (K-XF; 24°55′N, 120°58′E) is *Kandelia obovata*, while *Avicennia*

*marina* is the dominant species in Fangyuan (A-FY; 23°56′N, 120°19′E) and Beimen (A-BM; 23°17′N, 120°6′E). K-WZW and K-XF are situated in northern Taiwan, a subtropical region, with average annual precipitation values of 2023 and 1537 mm, respectively. A-FY and A-BM are located in southern Taiwan, a tropical region, with average annual precipitation values

of 1162 and 1603 mm, respectively. A-BM has the largest forest area (75.3 ha), while K-XF has the smallest (8.12 ha). Mean tree height across all sites ranged from 1.8 to 5.1 m, and tree density and diameter at breast height (DBH) averaged 0.6–2.4 tree m$^{-2}$ and 5.6–10.5 cm, respectively (Table 1). The tides were semidiurnal at all sites. The soil texture at all sites is silt, with an average grain size of 0.046 mm. During the summer (the study period), the average air temperature was 28.4 °C for *K. obovata* and 29.4 °C for *A. marina* (Lin et al., 2023). The sampling campaign was conducted from 1 June 2022 to 29 July 2022, with each site sampled for 3 days during the spring tide (Table 1). This period was chosen mainly because there is a higher GHG flux in summer compared to other seasons, as indicated by preliminary studies conducted at the same sites (Lin et al., 2020).

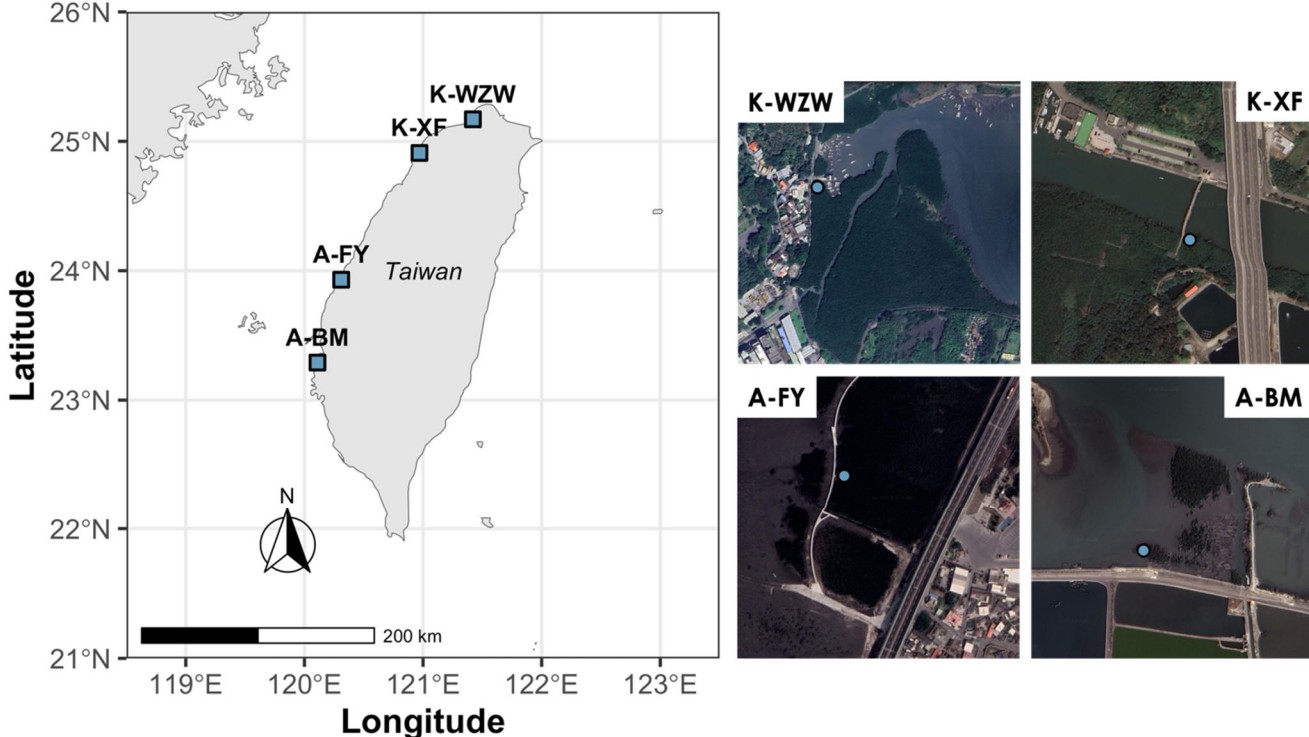

**Figure 1. Sample sites along the western coast of Taiwan. The blue dots represent the locations of sampling trees. K-WZW: Wazihwei; K-XF: Xinfeng; A-FY: Fangyuan; A-BM: Beimen. The dominant mangrove species in K-WZW and K-XF is *Kandelia obovata*, while *Avicennia marina* is the dominant species in A-FY and A-BM. Map sources: Natural Earth (left) and Google Earth (right).**

**2.2 Flux measurements**

At each sampling site, a mangrove tree was selected for the tree stem $CO_2$ and $CH_4$ flux measurements at approximately 110 cm above the ground. The specific height was chosen considering the potential maximum tidal height, which may reach up to

80 cm above the ground (Table 1). Due to the difference in the stem morphology, two distinct stem chambers—a semirigid chamber and a cylindrical chamber—were used in this study to measure the GHG emissions of *K. obovata* and *A. marina*, respectively (Fig. S1).

The semirigid chamber was modified from Siegenthaler et al. (2016) and was constructed from transparent recycled polyethylene terephthalate (rPET) bottles. A plastic sheet measuring 14 cm in length and 11 cm in width was cut from a bottle, and 2 cm wide and 1.5 cm thick chloroprene (CR) foam tape was attached around the edges and center of the plastic sheet, with two holes drilled and fitted with adapters for connecting the tubing, resulting in a chamber with a 16 $cm^2$ surface area and a 0.2 L volume. The chamber was installed on the tree stem with a strap prior to the measurement and subsequently removed. The second cylindrical chamber was constructed from a 0.2 L white polypropylene (PP) bottle, a 16 $cm^2$ square was cut from the lid, and two small holes were drilled at the bottom of the bottle; these holes were fitted with adapters to connect the tubing. The lid was fixed to the stem and sealed with silicone prior to the measurement. After each measurement, the chamber was removed, but the lid remained on the trunk (Fig. S1).

Two soil surfaces within 2 m of the sampled tree were selected for soil and water–atmosphere interface $CH_4$ and $CO_2$ flux measurements during the tidal cycle using a static chamber (Lee et al., 2011) and the floating chamber method (Lin et al., 2024), respectively. The soil chamber comprised a semicircular transparent polymethyl methacrylate (PMMA) cover (diameter of 30 cm) and a stainless steel ring (height of 16 cm and diameter of 30 cm) with an adapter on the cover for connecting the tubing. The ring was pressed into the soil before placing the cover over it, and a long-tailed clip was used to secure and cover the steel ring tightly to achieve an airtight seal (Fig. S1). During high tide, if the water level exceeded the height of the soil chamber (16 cm), the floating chamber was used (Fig. S1).

In this study, a portable gas analyzer (LI-7810, LI-COR Bioscience, NE, USA) was used to simultaneously measure $CO_2$ and $CH_4$ fluxes. The chamber was connected to the analyzer through tubing, and the gas inside the chamber was drawn into the analyzer with a pump, with each measurement lasting approximately five and seven minutes for the stem and soil, respectively. During the tidal cycle, stem and soil GHG fluxes were measured consistently. After each measurement was completed, the airtight sealed chamber was opened for approximately 3 minutes to allow the GHG concentration within the chamber to stabilize. The water level adjacent to the sampled trees was measured by a tape measure fixed on a PVC pipe (Fig. S1), simultaneously at the beginning of the flux measurement. To minimize soil disturbance, the researcher remained stationary at one location during the sampling campaign, avoiding walking around. Sampling was mainly conducted during daylight hours. Soil GHG flux data were mainly derived from Lin et al. (2024). The GHG flux (F) was calculated using the following equation:

$$F = (S \times V \times c)/(RT \times A) \qquad (1)$$

where S is the slope obtained from the linear regression of GHG concentration changes over time (ppb $CH_4$ $s^{-1}$; ppm $CO_2$ $s^{-1}$), V is the chamber volume (L), c is the conversion factor from seconds to hours, R is the ideal gas constant (0.082 L atm $K^{-1}$ $mol^{-1}$), T is the air temperature inside the chamber (K), and A is the surface area of the chamber ($m^2$). If the $R^2$ of the linear regression was < 0.7, the GHG flux was removed from the further statistical analysis. The surface area and volume of the semirigid chamber were calculated as described by Siegenthaler et al. (2016).

Different upscaling methods were applied to the tree stem GHG fluxes. First, the average fluxes during low and high tides were multiplied by the non-inundation time and inundation time length in hours, respectively. These values were then summed to calculate the daily fluxes, accounting for the tidal influence, which is denoted as "$F_{BothTide}$". Since sampling in mangrove forests was mostly conducted during low tide, the average fluxes during low tides were multiplied by 24 hours to scale up to daily fluxes, denoted as "$F_{LowTide}$", to compare with the fluxes accounted for tidal influence. The equations are shown below:

$$F_{BothTide} = (F_{high} \times t_{inundated}) + (F_{low} \times (24 - t_{inundated})) \tag{2}$$

$$F_{LowTide} = F_{low} \times 24 \tag{3}$$

where $F_{low}$ and $F_{high}$ are the average fluxes during low and high tides, respectively, $t_{inundated}$ is the average inundation time per day, acquired by multiplying the hours per day when the water level was higher than 0 cm by 2, since the tides are semidiurnal tides.

## 2.3 Statistical analysis

All the statistical analyses were performed in R 4.2.2 software. All the data were assessed for a normal distribution using the Shapiro–Wilk test. The Kruskal–Wallis test on ranks was used to evaluate the differences in the $CO_2$ and $CH_4$ fluxes between sites. To determine which sites differed, Dunn's multiple comparison test was applied as a post-hoc analysis when the differences were significant ($p < 0.05$). The relationships between the $CO_2$ and $CH_4$ fluxes during rising and falling tides were analyzed via a simple linear regression model. The results were considered statistically significant when the $p$ value was < 0.05. Data are primarily presented as the mean ± standard deviation (SD).

## 3 Results

During the study period, the mangrove tree stems served as net $CO_2$ sources, but there was distinct variation between sites (Fig. 2). In the *K. obovata* mangroves, the average $CO_2$ fluxes from the tree stems during the tidal cycle were $1.21 \pm 0.10$ mmol m$^{-2}$ h$^{-1}$ at the K-WZW site and $1.06 \pm 0.20$ mmol m$^{-2}$ h$^{-1}$ at the K-XF site (Fig. 2a). The stem $CO_2$ fluxes were significantly higher at the A-FY and A-BM sites, averaging $10.62 \pm 2.35$ mmol m$^{-2}$ h$^{-1}$ and $16.00 \pm 9.41$ mmol m$^{-2}$ h$^{-1}$, respectively (Fig. 2a). Across all sites, only the tree stem at the A-FY site functioned as a net $CH_4$ sink ($-0.17 \pm 0.52$ µmol m$^{-2}$ h$^{-1}$). However, the stem $CH_4$ fluxes at the K-WZW and K-XF sites showed no significant difference from the A-FY site, averaging $0.05 \pm 0.06$ µmol m$^{-2}$ h$^{-1}$ and $0.04 \pm 0.04$ µmol m$^{-2}$ h$^{-1}$, respectively (Fig. 2b). The stem $CH_4$ fluxes were significantly higher at the A-BM site ($0.48 \pm 1.17$ µmol m$^{-2}$ h$^{-1}$; Fig. 2b). Compared to those of the tree stems, the soils of the *K. obovata* and *A. marina* mangrove forests exhibited remarkably higher $CH_4$ fluxes, averaging $7.59 \pm 8.74$ µmol m$^{-2}$ h$^{-1}$ and $42.23 \pm 62.95$ µmol m$^{-2}$ h$^{-1}$, respectively. The average $CO_2$ flux from the soil was $1.73 \pm 2.31$ mmol m$^{-2}$ h$^{-1}$ in the *K. obovata* mangroves and $3.42 \pm 3.36$ mmol m$^{-2}$ h$^{-1}$ in the *A. marina* mangroves but did not differ significantly from that from the tree stems.

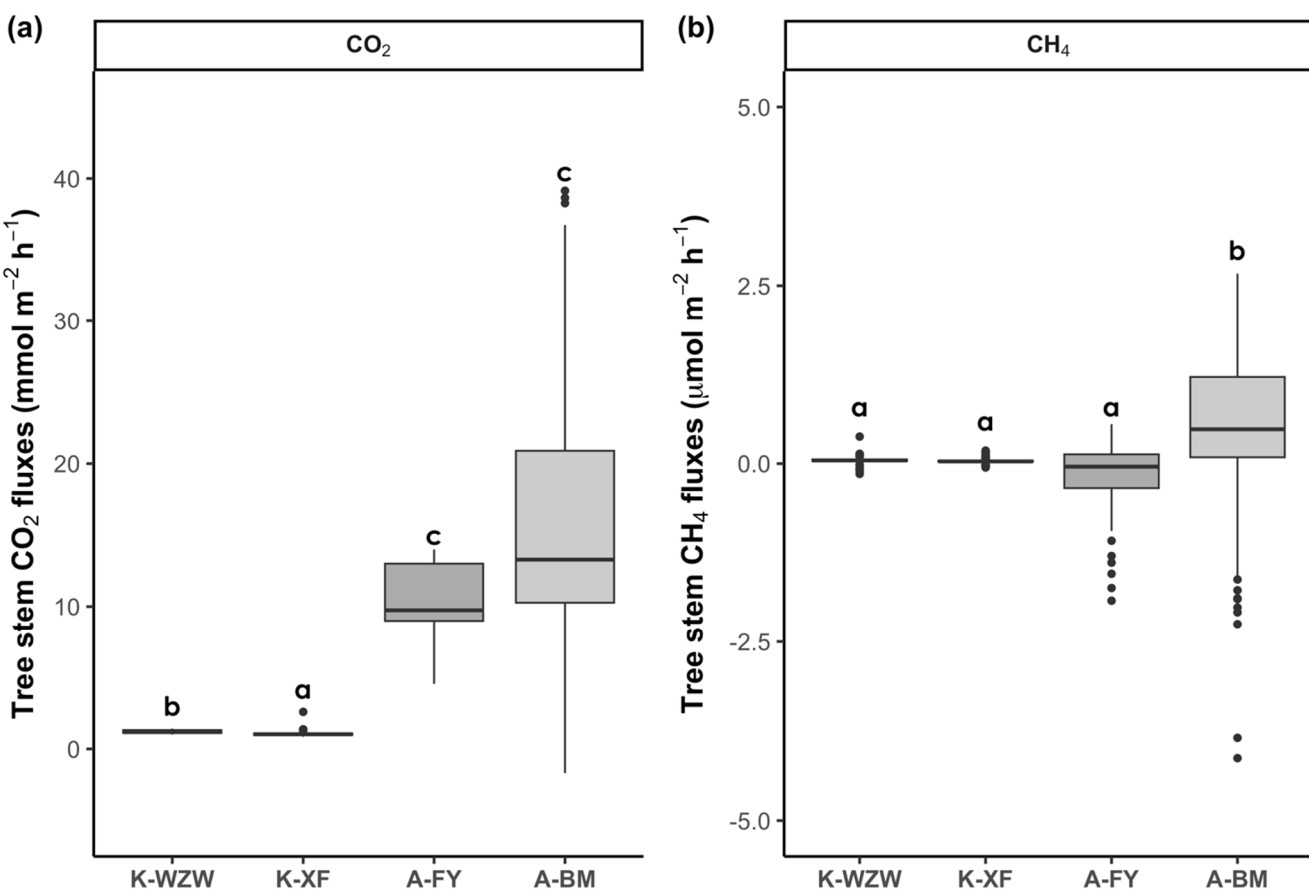

**Figure 2. Difference in the tree stem (a) CO$_2$ and (b) CH$_4$ fluxes among sites. Each data point represents a flux measurement during the tidal cycle (K-WZW: 88 replicates; K-XF: 82 replicates; A-FY: 75 replicates; A-BM: 152 replicates). Different letters above the boxplot indicate significant differences among sites, as determined by the Kruskal-Wallis test and Dunn's test ($p<0.05$).**

The mean inundation time and highest tidal height at each sampling site are provided in Table 1. During the tidal cycle, the

185 CO$_2$ fluxes from the mangrove tree stems exhibited different trends across all sampling sites (Fig. 3). The emissions remained relatively constant during the tidal cycle, ranging from 1.01 to 1.43 mmol m$^{-2}$ h$^{-1}$ and from 0.85 to 2.59 mmol m$^{-2}$ h$^{-1}$ at the K-WZW and K-XF sites, respectively (Fig. 3a). However, a sharp emission peak (2.59 mmol m$^{-2}$ h$^{-1}$) was observed at the K-XF site on Day 2 when the tide was falling, which was three-fold higher than the lowest flux (0.85 mmol m$^{-2}$ h$^{-1}$) measured on the same day (Fig. 3a). Similar to that at the K-WZW and K-XF sites, the CO$_2$ flux at the A-FY and A-BM sites generally showed

an increasing trend throughout the tidal cycle, ranging from 4.54 to 14.00 mmol m$^{-2}$ h$^{-1}$ and from -1.68 to 39.15 mmol m$^{-2}$ h$^{-1}$, respectively (Fig. 3a). However, this trend was observed at the A-FY site only on Day 1, when there was a distinct temporal trend in the increase in the CO$_2$ flux relative to that at the A-BM site. Specifically, the former started to increase before the flood current entered and stabilized after high tide, reaching a peak flux (10.36 mmol m$^{-2}$ h$^{-1}$) at the end of the measurement.

Conversely, the latter showed no significant change during the rising tide, followed by a steep rise toward high tide and a slight decrease during the falling tide; however, the $CO_2$ flux still remained higher than that during the preflood tide, ranging from -1.68 to 33.24 mmol $m^{-2}$ $h^{-1}$ during the rising tide and from 8.74 to 39.15 mmol $m^{-2}$ $h^{-1}$ during the falling tide (Fig. 3a).

**Table 1. Comparison of the upscaling methods with and without considering tidal influences on the $CO_2$ and $CH_4$ fluxes of mangroves.**

| | | K-WZW | K-XF | A-FY | A-BM |
|---|---|---|---|---|---|
| Dominant mangrove species | | *Kandelia obovata* | *Kandelia obovata* | *Avecinnia marina* | *Avecinnia marina* |
| Sampling date | | 2022-07-14/2022-07-16 | 2022-06-15/2022-06-17 | 2022-06-01/2022-06-02, 2022-06-18 | 2022-07-27/2022-07-29 |
| Sampling time | | 08:00/15:00 | 08:30/15:00 | 10:00/16:30 | 04:30/15:00 |
| Mean inundation time (h) | | 6.69 | 6.69 | 5.19 | 15.33 |
| Mean highest tidal height (cm) | | 58.1 | 70.5 | 47.3 | 77.5 |
| Flux measurement number (n) | | 88 | 82 | 75 | 152 |
| Stem $CO_2$ flux (mmol $m^{-2}$ $d^{-1}$) | $F_{BothTide}$ | 28.93 | 25.02 | 248.88 | 371.95 |
| | $F_{LowTide}$ | 28.94 | 24.82 | 245.95 | 339.99 |
| | Difference (%) | 0.03 | 0.81 | 1.19 | 9.40 |
| Stem $CH_4$ flux (µmol $m^{-2}$ $d^{-1}$) | $F_{BothTide}$ | 1.18 | 0.81 | -5.04 | 8.17 |
| | $F_{LowTide}$ | 1.22 | 0.76 | -5.49 | -0.74 |
| | Difference (%) | 3.68 | 6.21 | 8.35 | 1200.25 |
| Mean soil $CO_2$ flux (mmol $m^{-2}$ $d^{-1}$) | | 27.26 | 57.13 | 134.19 | 57.09 |
| Mean soil $CH_4$ flux (µmol $m^{-2}$ $d^{-1}$) | | 149.77 | 217.42 | 2404.28 | 345.37 |
| Mangrove forest area (ha) [a] | | 10.6 | 8.12 | 35.7 | 75.3 |
| Mean tree height (m) [a] | | 4.0 | 5.1 | 1.8 | 3.2 |
| Mean tree density (tree $m^{-2}$) [a] | | 1.3 | 2.4 | 1.0 | 0.6 |
| Mean diameter at breast height (cm) [a] | | 7.0 | 5.6 | 10.5 | 6.2 |
| Stem lenticel density (lenticels $cm^{-2}$) | | 0.08 | 0.05 | 1.83 | 2.96 |

$F_{BothTide}$: The average fluxes during low and high tides were added after multiplication with the corresponding time length. $F_{LowTide}$: The average fluxes during low tides were multiplied by 24 hours. The sampling date and time are in ISO 8601 format.
[a] The data was derived from Lin et al. (2021).

The $CO_2$ flux pattern observed during the tidal cycle differed between the tree stems and soils. Generally, the soil $CO_2$ flux peaked before and after high tide at all sites, either during the rising or falling tide, with the flood current just entering or leaving the sampling site (Fig. 3b).

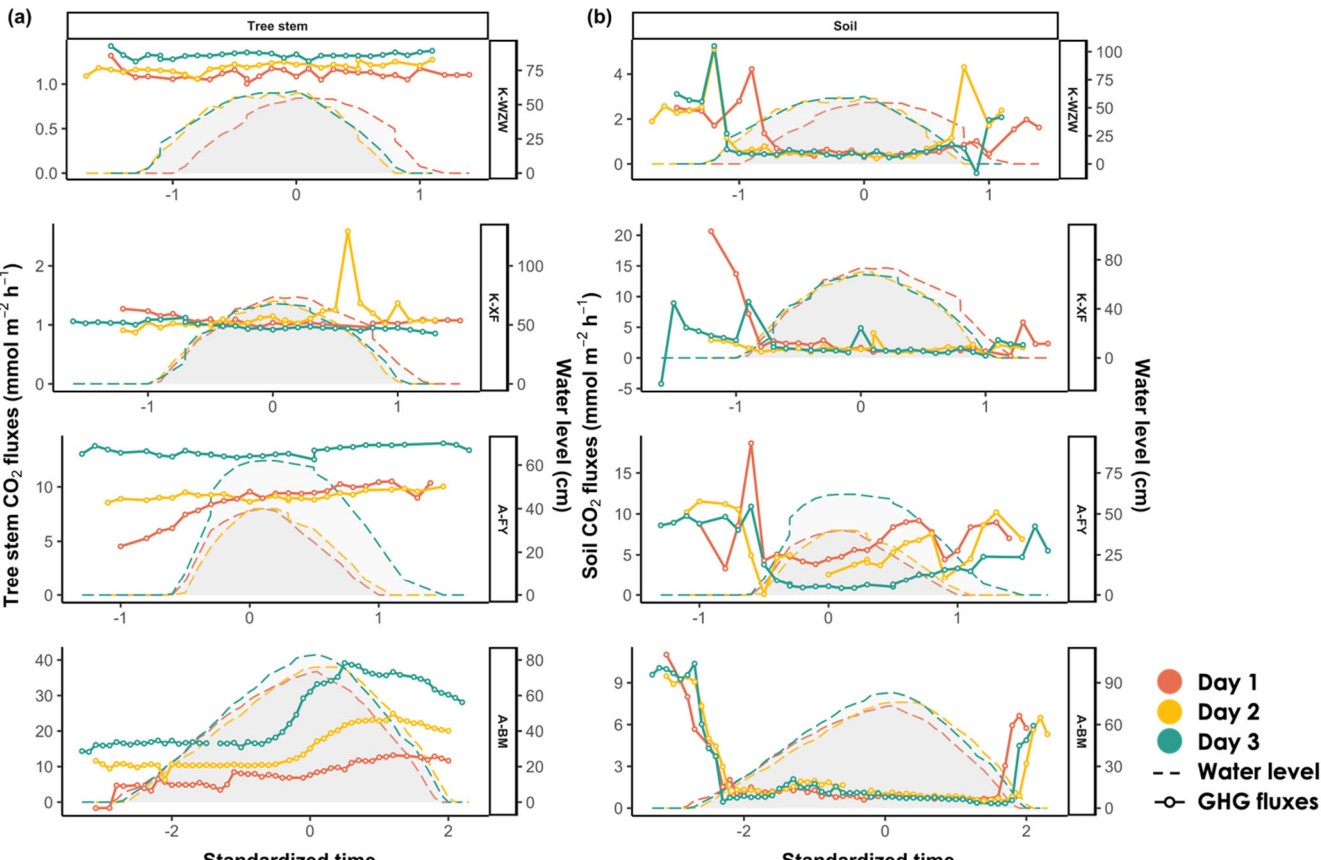

**Figure 3. Variations in (a) the tree stem $CO_2$ fluxes and (b) soil $CO_2$ fluxes during the tidal cycle. The time was standardized based on the time of the highest water level during the high-tide period (set as 0), then adjusted by decrementing the time by 0.1 for every 10-minute interval prior to the peak and incrementing by 0.1 for every 10-minute interval after the peak. The average values of the flux and water level were calculated when falling within each standardized time interval. The shaded area denotes the water level at the sampled tree. On Days 1, 2, and 3, the plant data were arranged chronologically according to the sampling date.**

Similar to those in the $CO_2$ flux, the $CH_4$ fluxes of *K. obovata* and *A. marina* exhibited distinct temporal trends during the tidal cycle (Fig. 4). In the *K. obovata* mangroves, there was significant variation in the stem $CH_4$ flux during the tidal cycle, ranging from -0.14 to 0.38 μmol m$^{-2}$ h$^{-1}$ and from -0.05 to 0.18 μmol m$^{-2}$ h$^{-1}$ at the K-WZW and K-XF sites, respectively, while consistent patterns were lacking between each sampling campaign (Fig. 4a). The stem $CH_4$ flux of *A. marina* increased throughout the tidal cycle, ranging from -1.92 to 0.55 μmol m$^{-2}$ h$^{-1}$ and from -4.13 to 2.67 μmol m$^{-2}$ h$^{-1}$ at the A-FY and A-BM sites, respectively. Specifically, the tree stems of *A. marina* functioned as $CH_4$ sinks before tidal inundation (A-FY: -0.53 ±

0.73 µmol m⁻² h⁻¹; A-BM: -0.64 ± 1.51 µmol m⁻² h⁻¹), but the CH₄ flux gradually increased thereafter, eventually becoming a CH₄ source during low tide (A-FY: 0.18 ± 0.24 µmol m⁻² h⁻¹; A-BM: 1.54 ± 0.56 µmol m⁻² h⁻¹). However, this pattern was not observed across all sampling campaigns (Fig. 4a).

For both mangrove species, the soil CH₄ flux during high tide (21.65 ± 45.29 µmol m⁻² h⁻¹) was lower than that during low tide (47.70 ± 63.27 µmol m⁻² h⁻¹) (Fig. 4b). Furthermore, there was a peak in the soil CH₄ flux during both tidal increase and decrease on all three sampling days, similar to the soil CO₂ flux (Fig. 3b; Fig. 4b).

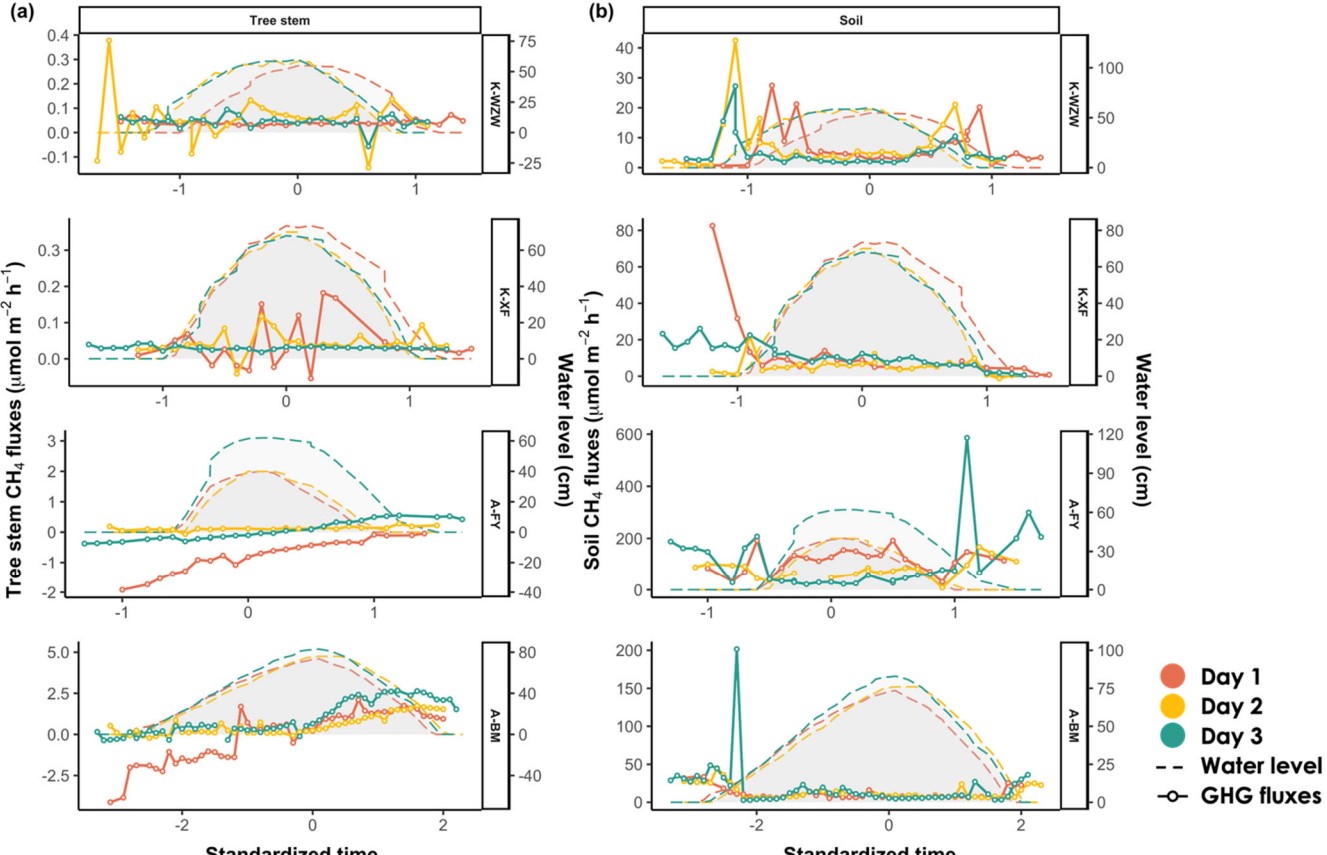

**Figure 4. Variations in (a) the tree stem CH₄ fluxes and (b) soil CH₄ fluxes during the tidal cycle. The time was standardized based on the time of the highest water level during the high-tide period (set as 0), then adjusted by decrementing the time by 0.1 for every 10-minute interval prior to the peak and incrementing by 0.1 for every 10-minute interval after the peak. The average values of the flux and water level were calculated when falling within each standardized time interval. The shaded area denotes the water level at the sampled tree. On Days 1, 2, and 3, the plant data were chronologically arranged according to the sampling date.**

During the tidal cycle, the CO₂ flux from the mangrove tree stems was positively correlated with the CH₄ flux during both the rising and falling tides. However, a significant relationship was detected only for *A. marina* (Fig. 5a; *p*<0.001). The CO₂ and CH₄ fluxes from both the stems and soils were simultaneously measured, and a negative correlation between the stem and soil

fluxes was observed across the two mangrove species. However, a significant relationship was detected only for *A. marina* during the falling tide (Fig. 5b, 5c; $p<0.001$).

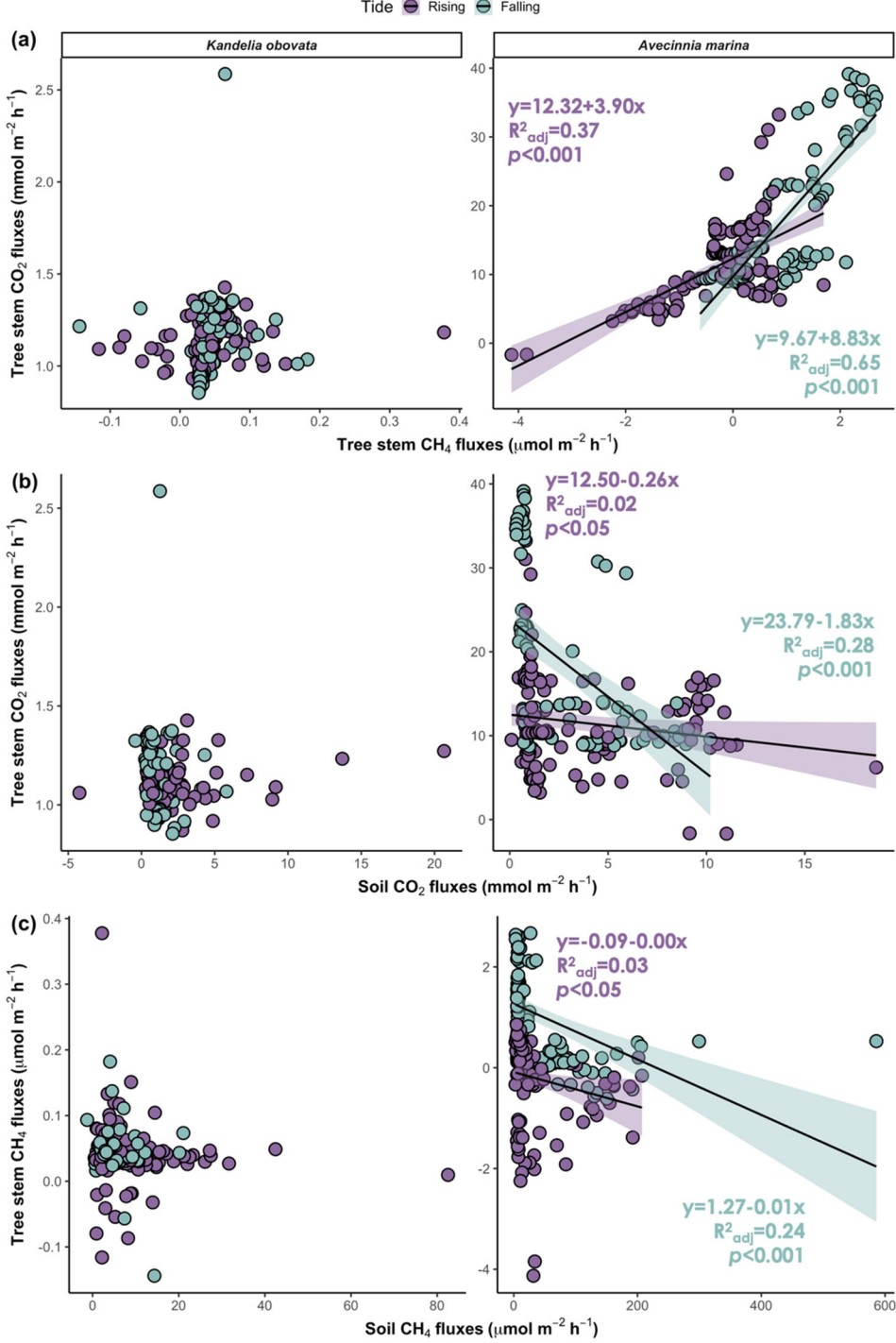

**Figure 5. Relationships between (a) the tree stem CO₂ and CH₄ fluxes, (b) tree stem CO₂ fluxes and soil CO₂ fluxes, and (c) tree stem CH₄ fluxes and soil CH₄ fluxes. The shaded areas denote the 95% confidence intervals of the regression lines.**

Since the tides at the sample sites were mainly semidiurnal tides, the average inundation time per day was calculated from the average time of high tide (when the water level was higher than 0 cm) during each sampling event multiplied by 2. The A-BM site exhibited the longest inundation time of 15.33 hours, while the inundation time during the sampling campaign was 6.69 hours at the K-WZW and K-XF sites and 5.19 hours at the A-FY site. The average highest tidal height (determined by the distance between the soil and water surface during high tide) was 58.1 cm at the K-WZW site, 70.5 cm at the K-XF site, 47.3 cm at the A-FY site, and 77.5 cm at the A-BM site. Different upscaling methods were applied to determine the tidal influence on the diurnal variation in the fluxes, where "$F_{BothTide}$" denotes the sum of the average fluxes during low and high tides after multiplication with the corresponding time length, and "$F_{LowTide}$" denotes the average flux during low tides multiplied by 24 hours. The GHG fluxes exhibited notable differences when tidal influences were considered (Table 1). Based on our findings, sampling only during low tide could underestimate the stem $CO_2$ and $CH_4$ fluxes on a diurnal scale, except at the K-WZW site, where the stem $CO_2$ and $CH_4$ fluxes were 0.03% and 3.68% lower when considering tidal influences (Table 1). At the K-XF, A-FY and A-BM sites, the differences in the stem $CO_2$ fluxes between the upscaling methods were smaller than those in the stem $CH_4$ fluxes, ranging from 0.81% to 9.40% (Table 1). The stem $CH_4$ fluxes at the K-XF site were approximately 6% higher when considering tidal influences, as opposed to ignoring tidal influences (Table 1). If the tidal influences were not accounted for, the mangrove tree stems at the A-FY and A-BM sites both acted as net $CH_4$ sink, while the $CH_4$ sink capacity was 8% and 1200% lower after accounting for tidal influences, turning the mangrove tree stem at the A-BM site into a net $CH_4$ source (Table 1).

## 4 Discussion

This study revealed distinct spatial and temporal variations in the $CO_2$ and $CH_4$ fluxes originating from tree stems and soils. Specifically, the sites dominated by *A. marina* exhibited up to 15 times higher $CO_2$ fluxes than sites dominated by *K. obovata*. The tree stems of *A. marina* at the A-FY site acted as a net $CH_4$ sink, while the A-BM site emitted $CH_4$ at approximately three times higher rate. In contrast, the tree stems of *K. obovata* at the K-WZW and K-XF sites were a weak $CH_4$ source compared to the tree stem at the A-BM site. The temporal dynamics during the tidal cycle also differed between the two mangrove species. Regarding *K. obovata*, the stem $CO_2$ and $CH_4$ fluxes at the K-WZW and K-XF sites lacked a consistent pattern between each sampling campaign. In contrast, *A. marina* exhibited an increasing trend in the $CO_2$ flux throughout the tidal cycle, whereas the $CH_4$ flux exhibited high temporal variability, functioning as a sink before tidal inundation and becoming a source during low tide at both A-FY and A-BM sites. Therefore, our results suggest that the different mangrove species, in this case, *K. obovata* and *A. marina*, may provide varying capacities for $CO_2$ and $CH_4$ exchange with the atmosphere through the tree stems during tidal cycles. Further investigation with a larger sample size needed to examine the hypothesis of mangrove species variation in GHG flux.

In terms of biological factors, *A. marina* contains pneumatophores, while *K. obovata* does not. Pneumatophores may facilitate the transport of oxygen to the rhizosphere and increase the oxidation–reduction potential, thereby inhibiting the methanogenesis process (Dušek et al., 2021). However, they can also serve as pathways for deep soil layer $CH_4$ emissions, facilitating $CH_4$ transport (He et al., 2019; Lin et al., 2021). In this study, pneumatophores were not intentionally avoided during the measurement. Therefore, the presence of pneumatophores may contribute to the increased soil $CH_4$ flux in the *A. marina* mangrove forest.

The GHG emissions of the stem, whether originating from the soil or the stem itself, require radial diffusion through the bark or lenticel to reach the atmosphere (Barba et al., 2019a). Radial diffusion is primarily influenced by biological factors such as wood density, wood moisture content, and lenticel density (Covey and Megonigal, 2019). A higher lenticel density, in particular, creates more pathways for GHG emissions, resulting in increased emissions (Zhang et al., 2022). Based on visual observation in situ, we found that the tree stems at the A-FY and A-BM sites exhibited a significantly higher lenticel density than those at the K-WZW and K-XF sites (Table 1). Therefore, it is speculated that the higher lenticel density of *A. marina* facilitates the emission of GHGs from the stem, resulting in a higher stem GHG flux at the A-FY and A-BM sites.

Previous studies on GHG emissions originating from mangrove tree stems were mostly conducted during low tide and under daylight conditions. Gao et al. (2021) showed that the stems of *Kandelia obovata* can both absorb and release $CH_4$, with average fluxes of -5.69 and 1.84 $\mu$mol m$^{-2}$ h$^{-1}$, respectively. Zhang et al. (2022) reported higher $CH_4$ emissions from *K. obovata* stems (7.04 $\mu$mol m$^{-2}$ h$^{-1}$), which dominated the ecosystem $CH_4$ flux of mangroves without pneumatophores. This contradicts the findings of this study, where the $CH_4$ emissions of *K. obovata* stems contributed less than the soil emissions. Liao et al. (2024) measured lower stem $CH_4$ fluxes from *K. obovata* during the winter (0.54 $\mu$mol m$^{-2}$ h$^{-1}$), which were 10 times higher than the average fluxes observed in this study. In the case of *A. marina*, the average stem $CH_4$ fluxes were 1.56 $\mu$mol m$^{-2}$ h$^{-1}$ (Jeffrey et al., 2019) and 2.79 $\mu$mol m$^{-2}$ h$^{-1}$ (Zhang et al., 2022) at the mangrove sites located in Australia and China, respectively. The tree stems of *A. marina* also exhibited $CH_4$ consumption capacity, with fluxes ranging from -33.96 to 48.83 $\mu$mol m$^{-2}$ h$^{-1}$, as reported in Gao et al. (2021). Regarding other mangrove species, *Kandelia candel* exhibited a stem $CH_4$ flux of -1.81 $\mu$mol m$^{-2}$ h$^{-1}$, while *Sonneratia apetala*, *Laguncularia racemosa*, and *Bruguiera gymnorhiza-Bruguiera sexangula*, which have the same specialized root structure as that of *A. marina*, provided stem $CH_4$ fluxes of 2.62, 0.87, and -0.49 $\mu$mol m$^{-2}$ h$^{-1}$, respectively (He et al., 2019). Epron et al. (2023) measured the $CH_4$ flux of the stems of *Bruguiera gymnorrhiza* throughout a 24-hour cycle, which ranged from -0.36 to 263.16 $\mu$mol m$^{-2}$ h$^{-1}$. In this study, the $CH_4$ fluxes of the stems of *A. marina* and *K. obovata* ranged from -0.14 to 0.38 $\mu$mol m$^{-2}$ h$^{-1}$ (K-WZW: 0.05 ± 0.06 $\mu$mol m$^{-2}$ h$^{-1}$; K-XF: 0.04 ± 0.04 $\mu$mol m$^{-2}$ h$^{-1}$) and from -4.13 to 2.67 $\mu$mol m$^{-2}$ h$^{-1}$ (A-FY: -0.17 ± 0.52 $\mu$mol m$^{-2}$ h$^{-1}$; A-BM: 0.48 ± 1.17 $\mu$mol m$^{-2}$ h$^{-1}$), respectively, which were at the low end of the reported range of stem $CH_4$ fluxes in previous studies (Table 2). Although $CH_4$ fluxes from mangrove tree stems generally decreased with increasing height (Epron et al., 2023; Gao et al., 2021; Jeffrey et al., 2019; Liao et al., 2024), average stem $CH_4$ fluxes of *A. marina* and *K. obovata* within similar heights to this study (> 1 m) were still higher. This may be due to site-specific variations in environmental conditions, tree physiology, and microbial activity, all of which can influence the production and consumption of methane by mangrove trees (Barba et al., 2019a; Covey and Megonigal,

2019). Further research is needed to delve into the underlying mechanisms which were not fully elucidated in this study due to limited data availability.

**Table 2. Comparision of tree stem methane (CH₄) flux in mangrove ecosystems reported in this study with previous literature. The values were presented as the range (minimum value–maximum value) and mean ± standard deviation.**

| Site | Period | Species | Height (m) | Stem CH$_4$ fluxes ($\mu$mol m$^{-2}$ h$^{-1}$) | Measurement technique | Reference |
|---|---|---|---|---|---|---|
| Australia | Winter (Aug 2018) | *A. marina* | 0.12 | 0.01–21.00 (4.03 ± 1.15) | CRDS | Jeffrey et al. (2019) |
| | | | 0.4 | 0.03–6.84 (1.21 ± 0.30) | | |
| | | | 0.8 | 0.31–4.77 (1.25 ± 0.19) | | |
| | | | 1.51 | 0.51–2.62 (1.14 ± 0.10) | | |
| China | All (Feb 2012– Nov 2013) | *L. racemosa* *S. apetala* *K. candel* *B. gymnorhiza-sexangula* | | 0.87 ± 0.81 2.61 ± 1.25 -1.81 ± 1.00 -0.49 ± 0.75 | GC | He et al. (2019) |
| | Summer (Jul 2019– Aug 2019) | *K. obovata* (Site 1) | 0.4 | -78.78–11.35 (-7.12) | CRDS | Gao et al. (2021) |
| | | | 1.4 | -52.67–8.89 (-4.39) | | |
| | | *K. obovata* (Site 2) | 0.4 | -32.36–26.90 (2.97) | | |
| | | | 1.4 | -9.95–51.38 (1.63) | | |
| | | *A. marina* | 0.4 | -33.96–22.50 | | |
| | | | 1.4 | -23.34–48.83 | | |
| | | *A. corniculatum* | 0.4 | -131.19–225.16 | | |
| | | | 1.4 | -41.42–42.43 | | |
| | Winter (Jan 2018), Summer (Jul 2018) | *K. obovata* *A. corniculatum* *A. marina* | 0–1.25 | (7.04 ± 3.96) (5.42 ± 3.04) (2.79 ± 2.13) | GC | Zhang et al. (2022) |
| | Winter (Dec 2021– Mar 2021) | *K. obovata* | 0.7 1.2 1.7 | (0.68 ± 0.17) (0.57 ± 0.19) (0.37 ± 0.13) | | Liao et al. (2024) |
| | | *S. apetala* | 0.7 1.2 1.7 | (1.25 ± 0.21) (0.84 ± 0.14) (0.42 ± 0.12) | | |
| Japan | Summer (July 2022) | *B. gymnorrhiza* | 0.3 | 1.80–825.12 (143.64) | | Epron et al. (2023) |
| | | | 0.6–1.5 | -0.36–263.16 (30.6) | | |
| Taiwan | Summer (Jun 2022– Jul 2022) | *K. obovata* (K-WZW) *K. obovata* (K-XF) *A. marina* | 1.1 | -0.14–0.38 (0.05 ± 0.06) -0.05–0.18 (0.04 ± 0.04) -1.92–0.55 | CEAS | This study |

| (A-FY) | (-0.17 ± 0.52) |
| *A. marina* | -4.13–2.67 |
| (A-BM) | (0.48 ± 1.17) |

**GC: Gas chromatography; CRDS: Cavity ring-down spectroscopy; CEAS: Cavity-enhanced absorption spectroscopy.**

The tree stem $CO_2$ and $CH_4$ fluxes exhibited similar temporal patterns during the tidal cycle. A significant positive relationship was also found between these fluxes, indicating that $CO_2$ and $CH_4$ emitted by mangrove tree stems may originate from the same source or be influenced by the same mechanism during the tidal cycle (Liao et al., 2024). According to previous studies, $CO_2$ emissions primarily occur through root respiration and stem respiration, as well as internal plant metabolism and transport from soils (Teskey et al., 2008). In contrast, $CH_4$ may be emitted or absorbed by methanogens and methanotrophs present in

tree bark or heartwood (Feng et al., 2022; Jeffrey et al., 2021). $CH_4$ emitted by tree stems may also originate from the soil, where the $CH_4$ produced in the soil enters the root system, enters the tree aerenchyma tissues or xylem, and is subsequently directly released into the atmosphere through the lenticel or tree stems (Barba et al., 2019a; Covey and Megonigal, 2019). Therefore, the fixation of $CO_2$, oxidation of $CH_4$, and emission of both GHGs by the tree stem may originate from the tree stem itself or from the soil. In this study, the transformation of tree stems from $CH_4$ sinks to $CH_4$ sources was observed in the

*A. marina* mangrove forest. This observation indicates that $CH_4$ emitted by tree stems may be affected by different sources during different periods of the tidal cycle.

The transport mechanism of GHGs in the stem is similar to that of herbaceous plants, occurring mainly by diffusion or evaporation, either jointly or individually. The diffusion direction mainly depends on the $CH_4$ concentration gradient. For example, if the gas concentration in the rhizosphere is high, GHGs can enter the plant root system either in gaseous or liquid

form, thus entering the aerenchyma or xylem tissue (Vroom et al., 2022). Aerenchyma is a specialized tissue found in many mangrove tree species (Evans, 2004). It comprises air-filled spaces that create gas transport pathways within the plant. Aerenchyma facilitates gas movement, including $CO_2$ and $CH_4$, within stems. Within the aerenchyma, $CO_2$ and $CH_4$ can diffuse or passively flow along concentration gradients. This transport pathway allows gases to move vertically within the plant, from the roots through the stem and ultimately into the atmosphere. Aerenchyma tissue is particularly important for $CH_4$ transport

because $CH_4$ is produced in oxygen-limited soils or in the rhizosphere by methanogens. The aerenchyma provides a direct pathway for $CH_4$ to move upward through the stems to be emitted into the atmosphere (Yáñez-Espinosa and Angeles, 2022). $CO_2$ and $CH_4$ can also dissolve during dilution and be transported within the xylem via sap flux (Takahashi et al., 2022). This study revealed the transition of mangrove tree stems from $CH_4$ sinks to $CH_4$ sources within the tidal cycle, which has not been observed in other studies, even with a high measurement frequency of upland tree stems at one-hour intervals (Barba et al.,

2019b). We speculate that the tree stem of *A. marina* may absorb $CH_4$ through the presence of methanotrophs during low tide (Jeffrey et al., 2021). During inundation, the diffusion of $CH_4$ produced in the deep soil layer may be restricted by the water pressure (Tong et al., 2013) since the pore spaces are filled with water. Tong et al. (2010) also reported a significantly lower $CH_4$ flux during inundation than during low tide. Therefore, we hypothesize that $CH_4$ produced in the soil during inundation periods may be primarily emitted into the atmosphere through tree stems (Vroom et al., 2022; Yáñez-Espinosa and Angeles,

2022) rather than being emitted across the water–atmosphere interface via diffusion or ebullition (Li et al., 2021), resulting in the observed gradual increase in the $CH_4$ flux throughout the tidal cycle. This hypothesis was also supported by the negative relationship between the soil and stem $CH_4$ fluxes of *A. marina* during both rising and falling tides observed in this study. However, the $CH_4$ flux of the tree stems of *Bruguiera gymnorrhiza* peaked after the tide receded (Epron et al., 2023), which does not support this hypothesis. It is critical to note that the specific mechanisms driving the observed peaks may vary depending on factors such as mangrove species, environmental conditions, tidal dynamics, and site-specific characteristics. However, further research is necessary to fully comprehend the underlying mechanisms.

To our knowledge, this study is the first to simultaneously measure the $CH_4$ fluxes of both stems and soils throughout the tidal cycle, even during tidal inundation. When quantifying the GHG emissions of mangrove tree stems, the discrete and continuous methods are two common measurement approaches. Discrete measurements involve sampling at specific time points with a lower temporal resolution but are practical and cost effective. Continuous measurements provide real-time monitoring with a high temporal resolution, accurately capturing short-term fluctuations and peak emissions but requiring specialized equipment and technical expertise. When considering tidal influences through continuous measurements, the $CH_4$ emitted by mangrove tree stems were significantly higher, with differences of up to 1200% for the stem $CH_4$ fluxes. Conversely, the upscaled $CH_4$ flux accounting for tides in tidal salt marshes was lower (Huang et al., 2019). When quantifying the GHG emissions of mangrove tree stems, discrete measurements are commonly used due to sampling difficulty at night and high tide. Although discrete measurements can still provide reliable estimates of the average emission rate over a specific period, they are useful only for broader-scale quantification and carbon and $CH_4$ budgeting models. This study highlights the need for continuous measurements of the GHG fluxes in coastal ecosystems, which can provide a more detailed understanding of emission patterns, aid in overall emission quantification, help individuals identify key drivers and mechanisms, reduce the uncertainty in GHG emissions, and facilitate the assessment of the impacts of specific events or environmental variables (Capooci and Vargas, 2022). However, when comparing practical, feasible, and cost-effective discrete measurements, continuous measurements require specialized equipment, technical expertise and intensive labor. It should also be noted that considerable differences were mainly observed at the A-BM site, with the longest inundation and highest water table.

This study provides insights into the potential tidal influence on GHG fluxes from mangrove tree stems. However, several uncertainties require further investigation. First, the study was conducted during the summer and daylight hours, which may have resulted in higher fluxes due to the effects of higher temperatures and the sap-flux dependent transport mechanism within the tree stems (Barba et al., 2019b; Köhn et al., 2021; Pangala et al., 2015; Pitz et al., 2018; Takahashi et al., 2022; Wang et al., 2016; Zhang et al., 2022). Second, the sampling campaign was conducted during spring tide, while $CH_4$ fluxes in tidal wetlands may differ between spring and neap tides (Huang et al., 2019; Tong et al., 2013). Third, sampling only at 110 cm height may have overlooked height-related GHG flux variations within mangrove tree stems, as observed in other studies (Epron et al., 2023; Jeffrey et al., 2019; Moldaschl et al., 2021; Pangala et al., 2013, 2014, 2015; Sjögersten et al., 2020). Finally, with the limited data availability, it is still uncertain whether there is a significant difference in GHG emissions from the tree stems between the two mangrove species..

**5 Conclusion**

This study revealed distinct temporal variations in the $CO_2$ and $CH_4$ fluxes of the tree stems of *A. marina* and *K. obovata* throughout the tidal cycles. While the GHG fluxes of *K. obovata* stems displayed inconsistent pattern, the $CH_4$ fluxes of *A. marina* stems suggesting a transition from a sink to a source, indicating the influence of different sources and mechanisms at different tidal phases. When considering tidal influences, the stem $CH_4$ flux could vary up to 1200% for *A. marina*, turning the stem from a net $CH_4$ sink to a source. This study highlights the need to consider tidal influences when quantifying the GHG

fluxes of mangrove tree stems and the potential limitations of discrete measurements relative to continuous measurements. However, further research is needed to fully understand the underlying mechanisms driving the observed flux variations and to improve our understanding and reduce the uncertainty in GHG dynamics in mangrove ecosystems.

*Data availability*. The original contributions presented in the study are included in the article. We encourage prospective data

users to contact us before embarking on any analysis.

*Author contributions*. Zhao-Jun Yong: Methodology, Investigation, Visualization, Writing - Original Draft. Wei-Jen Lin: Methodology, Investigation, Visualization, Chiao-Wen Lin: Methodology, Investigation, Visualization. Hsing-Juh Lin: Conceptualization, Supervision, Writing – Review & Editing, Funding acquisition.

*Competing interests*. None of the authors declare any conflict of interest.

*Acknowledgements*. This study was financially supported by the Ministry of Science and Technology of Taiwan under grant no. 112-2621-M-005-004 to HJL, and the "Innovation and Development Center of Sustainable Agriculture" from The Featured Areas Research Center Program within the framework of the Higher Education Sprout Project by the Ministry of Education (MOE) of Taiwan.

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
