# Peer review of "Tidal influence on carbon dioxide and methane fluxes from tree stems and soils in mangrove forests"

_EGUsphere, 2024_

## Author Response (AR1)

**Comments from RC 1**

**General comments**

*The manuscript by Yong et al investigated the drivers of mangrove tree stem methane can carbon dioxide fluxes over diurnal and tidal cycles. They also assessed soil fluxes and species differences from four mangrove sites, all located in Taiwan. This represents important work on the 'new research frontier' surrounding tree stem methane emissions, and in particular the role of mangrove stems as unaccounted-for CH$_4$ sources within blue carbon ecosystems, which undergo daily soil redox changes due to tidal and diel cycles, that complicate the extrapolation and drivers of tree stem emissions. Although the study captures high-resolution temporal data, a major weakness in the study design is only measuring one tree at one stem height, for each of the four sites. Because tree stem fluxes can have large variability between trees of the same stand/forest and even within axial tree stem heights of the same tree, the upscaled extrapolation and comparison between the sites is unfortunately weak due to this approach, so should not be a focus of the manuscript and either re-worked or removed. The strengths of the manuscript lie within making comparisons to drivers of tree stem fluxes (i.e. tidal influence and diurnal cycles) at each site, the shifts between uptake and emission of methane (on the same tree) and the high-resolution data coupled to WT, so should be the major focus.*

Response: Thanks for the valuable feedback on our manuscript. We appreciate your insights regarding the limitations of the study design and the need to enhance the robustness of our extrapolations. As you suggested, we have focused on highlighting the strengths of our research and addressed the limitations in the revision.

**Specific comments**

*Line 18 – for the abstract it may be useful to maybe state why you hypothesize sampling at low tide would overestimate stem fluxes.*

Response: Thanks for the reminder. Based on the results of the upscaled flux in this study, the statement has been reformulated for clarity as: " Based on our findings, the stem CH$_4$ fluxes of *A. marina* could vary by up to 1200% when considering tidal influence, compared to ignoring tidal influence. Therefore, sampling only during low tides might underestimate the stem CO$_2$ and CH$_4$ fluxes on a diurnal scale. " (L18-20)

*Line 51 – Seasonal water table height has also been shown to be one of the major factors driving wetland tree stem methane emissions with several studies over the past few years showing clear evidence for this, as this regulates the belowground soil oxygen, redox and methanogenesis conditions. This may be an important consideration for your study as focused on tidal WT fluctuations. See Terazawa 2021, Gauci 2021, Jeffrey 2023 for examples.*

Response: Thanks for the suggestion. The influence of water table height has been added in the revision as: "...which can be affected by the fluctuations of water table height due to seasonal changes and hydrological processes (Jeffrey et al., 2023; Peacock et al., 2024; Terazawa et al., 2021)." (L51-52)

*Line 62 – I would argue that mangrove soils are not a 'substantial' source of soil CH4 emissions and a reason Mangroves are considered ideal Blue Carbon sinks, compared to their freshwater counterparts - due to the abundant supply of sulphate from seawater (mentioned at line 76), favouring microbial sulphate reduction over methanotroph communities in mangrove sediments. Whilst true mangrove sediments can emit methane, it is often several orders of magnitude lower than freshwater wetland soils.*

Response: Thanks for the reminder. The statement primarily implies that the methane emitted from mangrove soil should not be overlooked when considering the carbon storage capacity of mangrove ecosystems (Rosentreter et al., 2021), while high methane emissions in coastal wetlands may still occur under high sulfate concentrations if the methanogenesis pathway was primarily methylotrophic methanogenesis (Capooci et al., 2024). However, we agree that it is indeed too aggressive to state that mangrove soil is a significant source of methane. Therefore, the term "substantial source" has been replaced with "source". (L66)

*Material and methods - seem to be missing information surrounding the forest stands at each site. Eg density of trees, tree height, average DBH and some background information about the four sites eg pristine vs anthropogenically altered or nutrient-enriched catchment systems. As this is a study of trees it is important to provide some of the basic forestry parameters used in upscaling, and any background site information which may biogeochemistry/ hydrology or nutrient inputs between sites. At line 228 'upscaling' is mentioned but there is no information as to how this was done and what caveats or assumptions this may contain so also needs to be mentioned. Details about any rejection threshold for poor linear fluxes (eg r2 values) should also be included to the methods. Some measurements had large methane uptake rates (very interesting) but it may also be useful to re- assess the starting concentrations of these incubation fluxes to ensure they are near atmospheric methane (1.8ppm) and did not include any accidental ebullition gas influencing the starting concentrations and the subsequent methane flux trend/gradient. CO2 flux is not always a useful proxy in those instances.*

Response: Thanks for the detailed feedback. We have updated the site description to include the missing information on each site's forestry parameters in Table 1. The upscaling methodology and rejection thresholds for low linear fluxes have been added in Section 2.2. While the initial concentration may vary at each site, we have made sure that it stabilized before each measurement by waiting approximately 3 minutes between each measurement. (L95-160)

*Flux measurement – The authors need to explain why they chose 110 cm stem height – as this may have actually represented a location of low methane emissions when compared to lower 10-50 cm stem closer to soils, as similar studies in mangroves and wetland forests have previously demonstrated. Was the tidal cycle expected to reach 1 m up the stem? This looks to be the case based on the SI photos! Also note how disturbance to soils was minimized during the intensive campaign measurements. Details about the volume and surface area of each chamber is also missing. The volume and SA of your chambers can affect the minimum detection limits of your equipment so suggest determining this value using the equations of Wassmann 2018 to be certain.*

Response: Yes, the specific height was chosen based on the potential highest tidal height,

which may reach up to 80 cm above the ground. To minimize soil disturbance, we tried to remain in one location during the sampling campaign, avoiding walking around. The method to calculate volume and surface area is from Siegenthaler et al. (2016), which was added in Section 2.2. (L115-160)

*Line 163 - How do the authors explain CO₂ uptake on tree stems?*
Response: The $CO_2$ uptake may be contributed by the $CO_2$ fixation by chloroplast-containing tissues in the tree stems (Teskey et al., 2008). This statement has been added in the Discussion. (L321)

*Line 227 – The authors mention 'diurnal' differences – were night time measurements collected? I may have missed this but could the data also be split into day vs night fluxes? Eg are there differences between night-time low tide vs daytime low tide – a period where tree transpiration, photosynthesis and root oxidation would differ? This data would also be novel and interesting to show instead of comparing the sites within the discussion. The discussion stating 'distinct' species variation in methane emissions would, unfortunately, require more than 1 tree replicate - as the variation may be due to the individual tree (age, size, physiological structure etc), the sampling date, the sampling site and soil differences as well.*
Response: Thanks for the reminder. The term "diurnal scale" was used since the upscaling method calculates the flux per day. Unfortunately, this study did not take nighttime measurements due to safety reasons.

*Conclusion – As mentioned above, I don't think you can conclude that one tree species 'distinctly' emits more methane than another, as only one tree was measured at each site. Stick to conclusions surrounding the drivers of stem fluxes, temporal variation range, and shifting between uptake vs emissions – which is quite interesting.*
Response: Thanks for the reminder. It is fully noted that measuring only one tree species at each site may limit the ability to determine a distinct difference between species conclusively. Therefore, the statement related to the "distinct difference" has been modified in the revision. (L382)

*Line 254-268 – the comparison to others studies are useful but further should discuss why this may be. Temperature, soil or at what stem heights were other comparative studies measured from? Would that explain larger emissions and lack of uptake measurements compared to your results? The data comparison with few other mangrove studies that exist could also be useful when presented in a comparison table.*
Response: Thanks for the suggestion. We have revised the manuscript by adding further discussion on the comparison of fluxes to other studies (L290-312). Additionally, we have incorporated a comparison table summarizing data from relevant studies to facilitate a clearer comparison and enhance the presentation of our findings. (Table 2)

**Technical corrections**

*Line 30 – change 'bubbles' to 'ebullition'*
Response: We have changed "bubbles" to "ebulliton". (L32)

*Line 32 – The Pangala 2017 would be a highly relevant citation here that considered top-down vs bottom-up scaling to compare tree stem contribution to methane budgets*

Response: Yes, we have already referenced this paper at the beginning of the sentence. Thanks for the suggestion. (L33)

*Line 50 – physiological bark-mediated gas transport on wetland trees was also recently shown as a pathway for stem methane.*
Response: Stem bark structure was included as one of the factors that influence the stem GHG fluxes, as demonstrated by Jeffrey et al. (2024). (L54)

*Line 55 – add: The contribution of 'mangrove' tree stems to the total...*
Response: We have changed "contribution of tree stems" to "contribution of mangrove tree stems". (L59)

*Line 69 – not sure N2O is relevant at all here as this is the only mention of this GHG in the entire paper. Stick to methane and CO2.*
Response: "and nitrous oxide (N2O)" was removed from the sentence. (L73)

*Line 117 – add: The 'second' cylindrical chamber was...*
Response: We have changed "the cylindrical chamber" to "the second cylindrical chamber". (L125)

*Line 163 – what is the +/-? SD or SE? please clarify throughout.*
Response: Data are primarily presented as the mean ± standard deviation (SD). The statement has been added in the Statistical analysis (L167).

*Table 1 . A row showing the number of fluxes n=? measured would be useful for readers as well as tree size. Some units are m2/d and others m2/h and switch between μmol and mmol. I suggest keep consistency throughout.*
Response: In Table 1, we have added a row showing the number of flux measurements, as well as the basic forestry parameters such as tree height, diameter at breast height (DBH), and others. The difference in units, specifically micromoles (μmol) and millimoles (mmol), was mainly due to the small value of methane ($CH_4$) compared to carbon dioxide ($CO_2$). The difference between fluxes represented as per day in Table 1 and per hour in other figures was because the fluxes in Table 1 were intentionally upscaled, as mentioned in Section 2.2.

*Figure 4 – a couple of large outliers for soil fluxes exist in the data. As per the comment about re-assessing your chamber start concentrations on tree stems, suggest double checking the start concentration on those soil chamber outliers in case ebullition was accidentally released into the chambers during measurements. However, these 'outliers' have also been shown to occur as natural fluxes (hot spots undergoing hot moments) when soil oxygen and methanotroph communities are limited by the fluctuating water tables etc.*
Response: In the field, we waited about 3 minutes to ensure the concentrations of both $CO_2$ and $CH_4$ returned to atmospheric levels before each measurement during sampling. This statement has been added in the Materials and methods (L139-141). The "outlier" was intentionally included to capture the potential tidal influence on the peak or drop in the greenhouse gas fluxes.

*Figure 5 – I would not show trend lines and equations for any non-significant data such as K. obovata. I also suggest the authors try other non-linear regression fits/ curves to the A.*

*marina data as some of the falling trends appear to be non-linear.*
Response: Thanks for the reminder. In this figure, regression analysis is primarily used to identify the relationship between variables rather than make predictions. Therefore, simple linear regression was used instead of more complex non-linear regression models, which may have higher R-squared values but could also lead to overfitting. The trend lines and equation of the non-significant regression were removed from Figure 5.

*Line 244- did you mean pneumatophores 'were' intentionally avoided?*
Response: No, we did mean the pneumatophores were not intentionally avoided during the measurements, which is why we also claim that the presence of pneumatophores may contribute to the increased $CH_4$ flux from the soil of the *A. marina* mangrove forest.

*Line 277 - change 'absorption' to 'oxidation' for methane. $CO_2$ would be 'fixation'.*
Response: The sentence had been changed to "the fixation of $CO_2$, oxidation of $CH_4$, and emission of both GHGs by the tree stem...". (L325)

*Line 281 – The net diffusion rate also relates to net oxidation rates during transport.*
Response: Thanks for the reminder. This sentence discussed the direction of diffusion rather than the net diffusion rate. We acknowledge that the net diffusion rate is the balance between the net oxidation and production rates of methane. (L329-330)

**Comments from RC 2**

*The authors provide useful insight into the carbon dioxide and methane fluxes in tree stems of mangrove trees and sediments. They also investigate the dial variation of the gas fluxes. They conducted chamber measurements to estimate the fluxes. The main findings of the manuscript are that tidal influence and mangrove tree species should be considered when quantifying the GHG fluxes. The text is clear. The figures need to be improved (fig 1, fig 3 and 4). The method needs to be improved, as it is not clear how the chamber measurement was conducted continuously in tidal cycles. Data of the paper should be uploaded in a public data repository.*
Response: Thank you for your insightful feedback on the manuscript. We acknowledge your comments regarding the clarity of the text and the need for improvements to Figures 1, 3, and 4. We have enhanced the method section to clarify how the chamber measurements were conducted continuously during tidal cycles. Regarding your suggestion to upload the data from the paper to a public data repository, we understand the importance of data accessibility and transparency in scientific research. However, in this specific case, we have decided to provide the data upon request (e.g., via email) due to related datasets that are currently being prepared for publication in a subsequent research paper. Your constructive criticism is invaluable in strengthening the manuscript.

**Introduction**

*Line 14: replace "markedly" to "remarkedly"*
Response: We have changed "markedly" to "remarkedly". (L15)

*Line 16,17 : reformulate the sentence*
Response: For clarity, the statement has been revised to " The stems of *A. marina* exhibited

an increasing trend in the $CO_2$ flux from low to high tides. On the other hand, $CH_4$ flux showed high temporal variability, with the stems of *A. marina* functioning as a $CH_4$ sink before tidal inundation and becoming a source after ebbing. " (L15-17)

*Line 65: not clear what is meant by "rising tide when tidal water reaches the sampling site".*
Response: It means the tidal phase when the water level starts to rise. The word "before rising tide" has been removed to avoid confusion. (L69)

*Line 81: Jeffrey et al. (2019) measured methane emissions from mangrove stems*
Response: Jeffrey et al. (2019) reported that dead mangrove trees may contribute approximately 26% to the $CH_4$ emissions in mangrove ecosystems. However, there is only one study on the GHG fluxes from mangrove tree stems during tidal cycles (Epron et al., 2023). (L84)

*Figure 1: Add a subplot on a zoom-in location for each site*
Response: Thanks for the suggestion. The subplots have been added in Figure 1.

**Method**

*Line 99: Is the experiment conducted during spring or neap tide?*
Response: The experiments were conducted during spring tides. The purpose is to capture the temporal variation of GHG fluxes over a longer time scale.

*Line 109:  Only 1 mangrove tree was selected? It is not representative.*
Response: Thanks for bringing this to our attention. According to the suggestion raised by Reviewer#1, this manuscript's strengths lie in making comparisons to drivers of tree stem fluxes at each site, the shifts between uptake and emission of methane on the same tree, and the high-resolution data coupled to the water table. We have focused on highlighting the strengths of our research and addressed the limitations in the revision of the last paragraph of Section 4.

*Lind 130: Is the stem measurement conducted continuously during tidal cycle? How many measurement was conducted if the measurement last for 5-7 minutes?*
Response: Yes, as mentioned in Section 2.2 (L115-150), the flux was measured consistently, with approximately 3-minute intervals, resulting in 23 to 54 measurements per day. This information has also been supplemented in Table 1.

*Line 132: Is there any nighttime measurements?*
Response. No nighttime measurements were taken due to safety concerns, as the greenhouse gas flux measurements were conducted manually. As stated in Section 2.2, "Sampling was mainly conducted during daylight hours." (L143)

*Line 135: State the relevant flux calculation and upscaled calculation (with and without tidal effect)*
Response: Thanks for the suggestion. The calculation processes have been added in Section 2.2. (L151-160)

*Line 132: Provide method that measure water level*
Response: Thanks for the suggestion. The method has been added in Section 2.2 as "The

water level adjacent to the sampled trees was measured by a tape measure fixed on a PVC pipe." (L141)

**Results**

*Line 141: For the comparison of species, were they present at the same mangrove site?*
Response: No, these species were not at the same site. In K-WZW and K-XF, the species are both *K. obovata*; as for A-BM and A-FY, the species are both *A. marina* (now stated in Table 1 as well).

*Table 1: state tidal range of each sites. Provide the soil fluxes.*
Response: Thanks for the reminder. "Highest tidal height" in Table 1 indicates the maximum water level recorded at each site throughout the sampling campaign. We have changed it to the "Mean highest tidal height", which shows the average of the highest water levels recorded at each site. This can also represent the tidal range, as the lowest water level recorded across all sites was 0 cm due to equipment restrictions. The mean soil fluxes at each site have been added to Table 1.

*Figure 3: It is a bit confused to have y axis (water level) from positive to negative. I suggest flipping the axis the other way around. For the data point, is there only 1 measurement for each data? As the chamber lasts for 5 minutes, how to capture continuous data and how many samples are included? Further specifications could be helpful. Consider renaming x axis name as it is confusing to have negative values. Also add the legend showing the different line type.*
Response: Thanks for the suggestion. The y-axis for water level has been flipped in both Figures 3 and 4. Each data point represents the average value of the measured fluxes that fall within the same standardized time interval; however, each time interval mostly contains one measurement. After each 5-minute measurement, the chamber was manually (for about 3 minutes) removed to stabilize the gas concentration within the chamber. In Section 2.2, the following specification has also been added: "During the tidal cycle, tree stems and soil GHG flux were measured consistently. After each measurement, the airtight sealed chamber was opened for approximately 3 minutes to stabilize the GHG concentration within the chamber." (L139-141) The samples included are now presented in Table 1. The x-axis from both Figures 3 and 4 has been renamed from "Time" to "Standardized time." The legend showing different line types has also been added.

*Line 193: Mention which sites have A.marina, as it is not indicated in the Fig 4*
Response: The sentence has been modified from "...tree stems of *A. marina* functioned as..." to "... the tree stems of *A. marina* functioned as $CH_4$ sinks before tidal inundation (A-FY: -0.53 ± 0.73 µmol m$^{-2}$ h$^{-1}$; A-BM: -0.64 ± 1.51 µmol m$^{-2}$ h$^{-1}$)...". (L221-223)

*Fig 5. Only plot lines with significant p values to avoid confusion*
Response: As Reviewer #1 also suggested, all the non-significant trend lines and equations have been removed.

*Line 225: I have concerns on "none" flux, as why only considering the low tide? State the reason. I suggest changing to "low tide", which indicates samples only taken during low tide.*
Response: Thanks for the reminder. Since the sampling in mangroves was conducted mostly during low tide, "none" flux represents the commonly used method to upscale GHG

flux in mangroves without considering tidal influence. The reason to calculate "none" flux is to compare it with the flux that considers tidal influence (which is the "tide" flux). The "tide" and "none" flux had been changed to $F_{BothTide}$ and $F_{LowTide}$ to avoid confusion. (L251-254)

**Discussion**

*In terms of $CO_2$, explain why there is a increasing trend in $CO_2$ flux from low and high tide?*
Response: The increasing trend in $CO_2$ flux may be due to the same mechanism that influences $CH_4$ flux during the tidal cycle. As Liao et al. (2024) indicated, distinct microbial groups involved in photorespiration might be present in the heartwood of *Kandelia obovata* and *Sonneratia apetala*, suggesting a connection between stem $CH_4$ flux and tree physiological activity, as well as microbial communities.

*Line 234: How many times higher ?*
Response: It is approximately 13 times higher for $CO_2$ and 7 times higher for $CH_4$. (L265-266)

*Compare $CH_4$ fluxes between tree stems with soil, why it is different?*
Response: Thanks for the suggestion. The results of this study support the hypothesis that the $CH_4$ fluxes emitted from tree stems originated from the soil. However, transporting $CH_4$ from the soil to the stem requires overcoming several barriers, which restrict the rate of $CH_4$ transport into and within the stems (Vroom et al., 2022). Additionally, the $CH_4$ transported within the stems may also be oxidized during the process (Jeffrey et al., 2021), ultimately leading to lower $CH_4$ fluxes from the tree stems than from the soil. (L336-349)

*Line 251: Any other literature supporting that A.marina had higher lenticel density*
Response: The stem lenticel densities within the same stem height interval (105–125 cm above ground) of *K. obovata* and *A. marina* located in Southeast China were 0.10 lenticels $cm^{-2}$ and 0.03 lenticels $cm^{-2}$, respectively (Zhang et al., 2022), which were opposite to our studies.

*Line 267: $CH_4$ sequestration?*
Response: The term sequestration was revised to consumption to indicate the oxidation of $CH_4$ potentially by the methane-oxidizing bacteria within the bark of *A. marina*.

*Line 300: is the negative relationship between soil and stem $CH_4$ occurred during rising or falling tide?*
Response: The negative relationship was observed during both rising and falling tides.

**Comments from CC 1**

I have had the privilege of reviewing the manuscript titled "Tidal Influence on Carbon Dioxide and Methane Fluxes from Tree Stems and Soils in Mangrove Forests" by Yong et al. The study presents a significant contribution to our understanding of greenhouse gas dynamics in mangrove ecosystems. The authors investigated methane emissions from mangrove tree stems and soil. The findings regarding carbon dioxide and methane fluxes

from Avicennia marina and Kandelia obovata tree stems offer valuable insights into species-specific dynamics. Additionally, the identification of temporal variability in methane fluxes, coupled with the implications for overestimation of fluxes when sampling solely during low tides, underscores the importance of considering tidal influence in quantifying greenhouse gas emissions accurately. Overall, I find this manuscript interesting and believe it enhances our understanding of mangrove ecology and the role of tidal dynamics in shaping greenhouse gas fluxes.

Response: We sincerely appreciate your positive feedback and insightful comments on our manuscript. Thanks for recognizing the significance of this study in enhancing the understanding of greenhouse gas dynamics in mangrove ecosystems. Your feedback is invaluable in guiding the improvement of our manuscript.

*However, I do have some comments and questions regarding the manuscript. First, I suggest the authors acknowledge the potential seasonal uncertainty. Because this study was only conducted in summer, the influence of tides on GHG emissions remains unclear in winter. It is plausible that there may not be a difference in the influence of tides on GHG emissions, given the authors' mention of higher GHG emissions in summer based on their previous study.*

Response: Thank you for the reminder. We acknowledge the limitations of the data, which was sampled only during the summer and daylight hours. Therefore, we have added statements addressing these limitations in the last paragraph of Section 4. (L371-380)

*Secondly, I find the experimental design somewhat unclear. Strengthening the explanation of the experimental design, particularly Lines 109-125 in Section 2.2, would help readers better understand the study's methodology and ensure transparency in the research process. It seems like the authors measure three stems on one tree and two soil spots at one site for 3 days. Thus, there are 2 to 3 spatial replicates and 3 temporal replicates. Do the authors only do the measurement on one tree at one site? Do the authors define measurements from the same tree species or from one site as replicates?*

Response: We are sorry for the confusion. At each of the four sites, one tree stem at 110 cm height and two soil surfaces were selected to measure greenhouse gas fluxes. Each was measured for 3 days. In Table 1, the replicates at each site, shown as flux measurement (n), represent the number of measurements on the same tree over the 3-day sampling campaign.

*Third, I am not sure how the authors define the tide and none tide periods as the definitions are found in the Table 1. I also suggest the authors to clarify whether or not the soil is inundated during the high tides when measuring the GHG fluxes. This may affect the GHG transportation in gaseous or liquid phases.*

Response: The calculation processes of the "Tide" and "None" upscaled fluxes are detailed in Section 2.2. Figures 3 and 4 show that the soil was inundated during high tides, as indicated by the water level exceeding 0 cm. This switch in the chamber was further explained in Section 2.2: "During high tide if the water level exceeded the height of the soil chamber (16 cm), the floating chamber was used (Fig. S1)."